# Caffeic Acid Phenethyl Ester Assisted by Reversible Electroporation—In Vitro Study on Human Melanoma Cells

**DOI:** 10.3390/pharmaceutics12050478

**Published:** 2020-05-24

**Authors:** Anna Choromanska, Jolanta Saczko, Julita Kulbacka

**Affiliations:** Department of Molecular and Cellular Biology, Wroclaw Medical University, 50-556 Wroclaw, Poland; jolanta.saczko@umed.wroc.pl (J.S.); julita.kulbacka@umed.wroc.pl (J.K.)

**Keywords:** caffeic acid phenethyl ester, melanoma, electroporation, oxidative stress, apoptosis

## Abstract

Melanoma is one of the most serious skin cancers. The incidence of this malignant skin lesion is continuing to increase worldwide. Melanoma is resistant to chemotherapeutic drugs and highly metastatic. Surgical resection can only be used to treat melanoma in the early stages, while chemotherapy is limited due to melanoma multi-drug resistance. The overexpression of glutathione *S*-transferase (GST) may have a critical role in this resistance. Caffeic acid phenethyl ester (CAPE) is a natural phenolic compound, which occurs in many plants. Previous studies demonstrated that CAPE suppresses the growth of melanoma cells and induces reactive oxygen species generation. It is also known that bioactivation of CAPE to its corresponding quinone metabolite by tyrosinase would lead to GST inhibition and selective melanoma cell death. We investigated the biochemical toxicity of CAPE in combination with microsecond electropermeabilization in two human melanoma cell lines. Our results indicate that electroporation of melanoma cells in the presence of CAPE induced high oxidative stress, which correlates with high cytotoxicity. Moreover, it can disrupt the metabolism of cancer cells by inducing apoptotic cell death. Electroporation of melanoma cells may be an efficient CAPE delivery system, enabling the application of this compound, while reducing its dose and exposure time.

## 1. Introduction

The use of natural substances as potential therapies is becoming more and more popular. They cause much fewer side effects than synthetic chemotherapeutics; however, in order to be used as standard drugs, their usefulness in the therapy of specific diseases should be confirmed [1]. Analyzing drug metabolic pathways is a material and necessary part of the drug unfolding process. Drug metabolism studies are very important, especially when the metabolites are pharmacologically active or toxic, or when the drug is widely metabolized [2].

Caffeic acid phenethyl ester (CAPE) is a natural bioactive polyphenol, which can be extracted from propolis [3]. Its biological activities are the result of the presence of hydroxyl groups within the catechol ring. CAPE is documented as having a wide range of medical applications. The literature presents its antioxidant, antimicrobial, anti-inflammatory, and cytotoxic properties [4,5,6,7,8]. It was observed that CAPE has antitumor activities [9] devoid of causing cytotoxicity to normal cells [10]. CAPE has a large potential against melanoma cells due to the specific metabolism of these cells. It was shown that CAPE is a selective glutathione *S*-transferase (GST) inhibitor in the presence of tyrosinase, which is abundant in melanoma cells. CAPE is metabolized by tyrosinase to CAPE-SG conjugate (caffeic acid glutathione conjugate) and CAPE-quinone, which inhibit GST catalytic activity and play major roles in the selective inhibition of GST in melanoma cells [11].

Saleem et al. postulated that the cytotoxicity of CAPE is related to its apoptotic effect [12]. Liao et al. carried out a cytotoxicity study of CAPE in colon adenocarcinoma cells (CT26) and showed a dose-dependent decline in cell viability. There was also an observed reduction in the expression of matrix metalloproteinase and the production of vascular endothelial growth factor in CAPE-treated CT26 cells. This causes a reduction in angiogenesis and metastasis [13]. It was proven that CAPE is capable of blocking nucleotide production [11], obstructing the tumor promoter-mediated oxidative responses in the culture of cancer cells [14], and inhibiting the colonic preneoplastic lesions and enzymatic processes related to colon carcinogenesis [13].

Kudugunti et al. studied the effects of CAPE on human melanoma cell lines. They observed that CAPE was selectively toxic toward five melanoma cell lines in comparison to four non-melanoma cell lines. They suggest that CAPE is metabolized by tyrosinase to its quinone form, and this conjugate played a major role in the inhibition of glutathione *S*-transferase (GST) [15]. GST is an important enzyme in the detoxification of a broad range of compounds, with a vital role in the glutathionylation of cellular proteins in cancer. Overexpression of GST promotes the multi-drug resistance of cancer cells [16]. In a previous study of GST expression, it was shown that GST is highly expressed in melanoma cells when compared to normal cells [17]. However, none of the previous studies investigated the effectiveness of CAPE toxicity when CAPE transport into cells is supported by cell membrane electroporation (EP). The membrane permeability is an important factor in drug absorption. Gou et al.’s studies showed that CAPE transport across the cell membrane is limited, which is likely due to the *p*-glycoprotein overexpression [18]. It is the main efflux pump for a variety of chemicals, which play an important role in the absorption and distribution of many polyphenolic compounds, including CAPE [19]. Malignant melanoma shows high levels of intrinsic drug resistance associated with a highly invasive phenotype, which correlates with *p*-glycoprotein overexpression [20]. For this reason, we decided to increase the efficiency of CAPE delivery to melanoma cells using reversible electroporation. Electroporation means the use of short high-voltage pulses to overcome the natural barrier of the cell membrane. By applying an external electric field, a reversible breakdown of the membrane can be induced. This transient state can be used to load cells with a variety of different molecules. Initially, EP was used for gene transfer; now, it is in use for delivering a whole range of different molecules, such as drugs, dyes, antibodies, and oligonucleotides. EP was proven useful for in vitro and in vivo studies, as well as in patients, where it is used for drug delivery to malignant tumors [21].

Our study aimed to investigate and compare the effectiveness of CAPE and CAPE connected with EP against melanoma cells. A cell viability assay was performed to assess the cytotoxicity of CAPE toward melanoma cells. The implications of intracellular glutathione (GSH) depletion, lipid peroxidation, and induction of apoptosis were also studied.

## 2. Materials and Methods

### 2.1. Cell Culture

Two human melanoma cell lines were tested—the primary cell line (MeWo) and that derived from a metastatic region (Me45). The MeWo cell line (BRAF^WT^ melanoma cell line) was obtained from ATCC^®^ (LGC Standards, Lomianki, Poland). The Me45 cell line (BRAF^V600E^ melanoma cell line) obtained from a lymph node metastasis of skin melanoma was a gift from the Radiobiology Department of the Center of Oncology in Gliwice, Poland. The cells were cultured in Dulbecco’s modified Eagle medium (DMEM, Sigma-Aldrich, Poznan, Poland), supplemented by 2 mM l-glutamine and 10% fetal bovine serum (FBS, Sigma-Aldrich, Poznan, Poland). Cells were cultured at 37 °C in 5% CO_2_. For cell passage, cells were removed by 0.25% trypsin with EDTA (Sigma-Aldrich, Poznan, Poland).

### 2.2. Electroporation

Electroporation was performed using an ECM 830 Square Wave Electroporation System (BTX Harvard Apparatus, Syngen, Biotech, Wroclaw, Poland). During electroporation, procedure cells were suspended in buffer with low electrical conductivity (10 mM KH_2_PO_4_/K_2_HPO_4_, 1 mM MgCl_2_, 250 mM sucrose, pH 7.4). After trypsinization and centrifugation (5 min, 1000 rpm), cells were counted (cells density = 3 × 10^6^ cells/mL) and resuspended in 200 µL of EP buffer or EP buffer with CAPE. The electroporation protocol was chosen based on previous studies [22,23] in which non-toxic electroporation parameters were established for Me45 and MeWo cell lines. These parameters were as follows: eight pulses with 100-µs pulse duration with a 1-s interval; electrical fields of 1000 V/cm. The cell suspension was pulsed in a cuvette with two aluminum plate electrodes. After pulsation, cells were left for 10 min with the addition of DMEM at 37 °C; then, they were centrifuged and subjected to analyses.

### 2.3. Cellular Viability—MTT Assay and IC_50_ Determination

The cell viability was estimated using a cellular metabolic activity test—MTT assay (Sigma-Aldrich, Poland). This colorimetric test is based on the reduction of tetrazolium salt to formazan crystals by metabolically active mitochondria. The insoluble formazan crystals were dissolved using isopropanol with the addition of hydrochloric acid. An obtained colored solution was quantified by measuring absorbance at 570 nm using a multi-well spectrophotometer (EnSpire, Perkin Elmer, Krakow, Poland). The mitochondrial metabolic function was represented as a percentage of viable treated cells with respect to untreated cells. Caffeic acid phenethyl was purchased from Sigma-Aldrich (Sigma-Aldrich, Poznan, Poland). As a solvent for CAPE, dimethyl sulfoxide (DMSO, Sigma-Aldrich, Poznan, Poland) was used. The DMSO concentration in the samples with the highest CAPE solution was 1.2%. CAPE was used in the experiments at concentrations ranging from 1–50 µM. The MTT test was performed on 96-well plates, where 1 × 10^4^ cells were seeded for one well. The estimation of viability was carried out after 24 and 72 h. All samples were analyzed six times. Based on the viability values obtained in the MTT assay, the IC_50_ values for CAPE and CAPE combined with EP were calculated using Quest Graph™ IC_50_ Calculator (AAT Bioquest, Inc, Sunnyvale, California, CA, USA). IC_50_ was determined with a non-linear model.

### 2.4. Cloning Efficacy Test

Cells were seeded onto culture dishes (250 cells/dish, 35 mm in diameter) for 11 days to obtain macroscopically visible colonies. After 11 days, maintained in a humidified atmosphere at 37 °C, the colonies were fixed with 5% formaldehyde, stained with 1% crystal violet, and counted. The number of colonies was expressed as the percentage of viable treated cells relative to untreated control cells (without CAPE and EP). All samples were analyzed six times.

### 2.5. GSH/GSSG Assay

The GSH/GSSG ratio was measured using a luminescence-based system that detects and quantifies total glutathione (GSH + GSSG) and its oxidized form (GSSG) in cultured cells (GSH/GSSG-Glo™ Assay, Promega, Perkin Elmer, Poland). Reduced (GSH) and oxidized glutathione both subsist in healthy cells, but a change in GSH and GSSG levels is essential when evaluating oxidative stress, potentially leading to apoptosis or cell death. Determination of the total and oxidized form of glutathione was performed in parallel reactions. In one configuration, the assay reagents measured total glutathione using a reducing agent. In a second configuration, the assay reagents were used to measure only the oxidized form. Because the assay was conducted directly on cells in culture wells, loss of GSH or GSSG was minimized. After treatment with CAPE or CAPE + EP, cells were seeded into white 96-well microtiter plates at the concentration of 5 × 10^3^ cells/well. The level of reduced and oxidized glutathione (GSH/GSSG) was determined after 24 h using a multi-well scanning plate reader (EnSpire Perkin Elmer, Krakow, Poland). All samples were analyzed six times.

### 2.6. Lipid Peroxidation

MDA is the most commonly used biomarker of oxidative stress in the cell. MDA is generated during the reaction from the decay of products of lipid peroxidation. The assay with TBA was based on a compression reaction of two molecules of TBA with one molecule of MDA. The reaction rate depended on the concentration of TBA, temperature, and pH. The reaction was taken out in acidic solution at a temperature of about 100 °C. The final product (MDA reacted with TBA to form a colored complex) was measured by the absorbance at a wavelength of 535 nm (EnSpire Perkin Elmer, Krakow, Poland). The level of MDA was determined after 24-h treatment with CAPE or CAPE + EP. All samples were analyzed six times.

### 2.7. Immunocytochemical Cleaved PARP-1 Protein Evaluation

Immunocytochemistry was performed after 6 h of incubation using the avidin–biotin complex (ABC) method. This is a standard and one of the most extensively used techniques for immunocytochemistry staining. Avidin, which is a large glycoprotein, can be labeled with peroxidase and has a very high affinity for biotin. This low-molecular-weight vitamin is used in that method to form a conjugation with antibodies. The immunohistochemical staining intensity is a function of the enzyme activity, and enhanced sensitivity can be obtained by expanding the number of enzyme molecules bound to the antigen. The multiple binding opportunities between the tetravalent avidin and biotinylated antibodies (connected to the antigen) allow obtaining clear marked antigen–antibody binding sites. The procedure involved the following stages, based on applying three layers: the first layer was an unlabeled primary antibody; the second layer was the biotinylated secondary antibody; the layer one was a complex of avidin–biotin–peroxidase. The peroxidase then interacted with diaminobenzidine (DAB), which resulted in creating colorimetric end products. Fixed melanoma cells after CAPE or CAPE + EP treatment were stained. The procedure was performed on basic microscope slides. Anti-cleaved PARP1 antibody (Abcam, ab32561, Cambridge, UK) was used as the primary antibody using a 1:100 dilution. Other reagents were from the staining kit (Dako REAL EnVision Detection System, Peroxidase/DAB+, Rabbit/Mouse, K5007, Agilent, Santa Clara, CA, USA). Cell nuclei were stained with hematoxylin (Roth, Karlsruhe, Germany). The microscope slides were examined with an upright microscope (Olympus BX51, Tokyo, Japan). Staining assessments were determined by counting 100 cells in randomly chosen fields. The staining was estimated positive if it was discerned in more than 5% of cells. The intensity of staining was assessed as (−) negative or (+) weak, (++) moderate, and (+++) strong.

### 2.8. Apoptosis and Necrosis Evaluation

Annexin V conjugated to the FITC fluorochrome was used for flow cytometric analysis of cells undergoing apoptosis. The externalization of phosphatidylserine occurs in the early stages of apoptosis, and FITC annexin V staining can identify apoptosis at an earlier stage than assays based on nuclear changes such as DNA fragmentation. Necrotic changes were identified using propidium iodide (PI), in which fluorescence is enhanced by 20–30-fold upon binding to nucleic acids. The analysis was carried out using a commercial FITC annexin V kit (BioLegend, San Diego, CA, USA, catalog no. 640914). For apoptosis and necrosis evaluation, 10, 25, and 50 μM CAPE was used. The analysis was performed on a CyFlow Cube 6 cytometer (Sysmex, Europe GmbH, Norderstedt, Germany), where the FL-1 detector was used for annexin-V–FITC and the FL-2 detector was used for necrotic cells (PI-stained) measurements. The analysis was performed after 6-h incubation with CAPE or after 6-h incubation of EP with CAPE. All samples were analyzed in triplicate; for each sample, a minimum of 10,000 events were analyzed.

### 2.9. Statistical Analysis

The results were analyzed statistically with GraphPad Prism 7.03. For statistical analysis, a two-way ANOVA test was used with Dunnett’s multiple comparisons test, which compares all treated samples to control samples not exposed to any treatment. In the second statistical test—Sidak’s multiple comparisons test—the effect of different concentrations of CAPE was compared to all respective doses of CAPE connected to EP.

## 3. Results and Discussion

There are many investigations in the literature that explored the biological activity of CAPE. This includes the anticancer properties of this compound. Our results obtained from the viability test showed a decrease in melanoma cell viability with increasing CAPE concentration. In both cell lines, EP application significantly increased the cytotoxic effect compared to samples without EP. After 24-h incubation, cell viability was lower than 30% for the highest CAPE tested concentration, where the transport of CAPE was assisted by electroporation (Figure 1A). After a longer incubation time, the tendency was the same, but the cytotoxic effect was stronger. Metastatic melanoma cells (Me45 cells) were more sensitive than primary melanoma cells (MeWo) (Figure 1B). Dunnett’s multiple comparisons tests showed statistically significant differences in cell viability after CAPE and CAPE with EP treatment compared to control samples not exposed to any treatment (Figure 1).

This might be presumably due to different tyrosinase levels in tested cells. Wadhawa et al. examined the cytotoxicity and growth inhibition of CAPE and CAPE in combination with γ-cyclodextrin by MTT assay in a variety of human cancer cell lines, including a melanoma cell line (G361). The results were similar to ours, but they enhanced the antitumor potency of CAPE using γ-cyclodextrin. The viability of melanoma cells was decreased after selected concentrations of CAPE [24]. A combination of electroporation and CAPE application efficiently reduces the time required to induce a cytotoxic effect in melanoma cells. Our data showed that the treatment of electroporation with CAPE on melanoma cells is more effective in reducing melanoma cell viability than CAPE treatment only. Moreover, the metastatic melanoma cell line (Me45) was more sensitive than primary melanoma cells (MeWo). Electroporation enhanced the toxicity effect of CAPE and, at the same time, allowed the selective action of this ester. The IC_50_ values of CAPE for Me45 and MeWo cells were 86 µM and 106 µM, respectively, while those for samples treated with CAPE combined with EP were 28 µM and 70.7 µM, respectively. The results obtained in the MTT test were also confirmed by clonogenic assay which is based on the usage of a definite number of cells in the test (cloning efficacy test) (Figure 2). After 11 days of incubation, viability of the Me45 cell line was 39% and 18% in samples treated with 50 µM CAPE in and 50 µM CAPE connected with EP, respectively. For MeWo, cell viability in these conditions was 60% and 39%, respectively. Dunnett’s multiple comparisons tests showed statistically significant differences in cell viability after CAPE and CAPE + EP treatment compared to control samples not exposed to any treatment (Figure 2).

Other results showed that CAPE activity, without electroporation, selectively destroys the cancerous cells, leaving normal, physiological cells, as observed in human immortal lung fibroblast WI-38 cells [25]. Reactive oxygen species generated by lipid peroxidation occur near cellular membranes and produce other free radicals, which modify proteomic and genomic processes. There are already available data that indicate the antioxidant properties of CAPE [26,27]. On the other hand, Kudugunti et al. observed a significant increase in ROS formation when melanoma cells were co-incubated with CAPE. They investigated the rate of ascorbic acid and NADH depletion to measure the extent of CAPE oxidation by tyrosinase/O_2_ and HRP/H_2_O_2_. It is known that peroxidase catalyzes the one-electron oxidation of catechols, while tyrosinase oxidizes catechols through a two-electron oxidation mechanism. Their data suggested that CAPE was more rapidly oxidized by HRP/H_2_O_2_ than tyrosinase/O_2_, which suggested an important role for the semi-quinone pathway in melanoma cell toxicity [28].

Glutathione conjugation plays an important role in defending against lipid peroxidation products. When this system is insufficient, cells are directed to the death pathway. Oxidized glutathione is an indicator of cell health and oxidative stress. A significant decrease in the ratio of glutathione’s reduced form to oxidized form was observed in the samples after CAPE and EP application in both tested melanoma cell lines. This ratio decreased by 86% and by 73% compared to the control cells for the Me45 and MeWo lines, respectively. Dunnett’s multiple comparisons tests showed statistically significant differences in the ratio of reduced and oxidized glutathione (GSH/GSSG) after CAPE and CAPE with EP treatment compared to control samples not exposed to any treatment (Figure 3).

A marked, statistically significant increase in malondialdehyde (MDA) levels was observed in melanoma cells treated with CAPE in combination with EP (Figure 4).

Kudugunti et al. investigated five melanoma cell lines and demonstrated that CAPE has an anti-proliferative effect and it is involved in ROS formation and intracellular GSH depletion in the presence of tyrosinase, which is abundant in melanoma cells [15]. Tyrosinase is an enzyme in the melanin synthetic pathway, and its expression is only found in melanin-producing cells [29]. Tyrosinase in melanoma cells transforms CAPE into a complex with glutathione called quinone. This conjugate inhibits glutathione *S*-transferase, which plays an important role in multidrug resistance, and it is overexpressed in cancer cells, as well as in melanoma [15,30]. Moreover, analysis through molecular modeling showed that CAPE binds to the catalytic active site of GST and promotes the formation of quinones. These conjugates can play a significant role in the selective inhibition of GST in SK-MEL-28 melanoma cells [15]. Other studies indicated that CAPE selectively caused an increase in the ROS formation and intracellular GSH decrease in various primary and metastasized melanomas, which express tyrosinase [28]. The overproduction of ROS which is manifested by lipid peroxidation, as well as protein and DNA damage, can lead to cell death [14]. We examined cleaved PARP (fragment of the PARP (p85) protein) to confirm the apoptotic pathway in examined cells. PARP-1 is a cellular substrate of caspases. Cleavage of PARP-1 by caspases indicates the induction of apoptosis. A high level of cleaved PARP-1 protein was observed in samples treated with CAPE and CAPE combined with EP, which may indicate induction of the apoptotic pathway (Figure 5, Table 1).

The immunocytochemical analysis also showed that the highest dose of CAPE in combination with EP was highly cytotoxic because only single cells were visible in the microscopic images (Figure 4). The presence of cleaved caspase-3 and cleaved PARP protein after cell incubation with CAPE was also observed in other previous studies [25,31,32,33]. However, the expression of cleaved PARP in the Me45 cell line was already observed after 10 μM CAPE exposure with and without electroporation, while, in MeWo cells, the expression of cleaved PARP was only noted for 10 μM CAPE assisted by electroporation (Figure 5, Table 1). We suggest that applying electroporation allows reducing the cytotoxic dose and exposure time of CAPE on examined cells. Flow cytometry analysis showed a clear induction of apoptosis after a 6-h incubation of cells with CAPE or CAPE combined with EP. However, it was observed that CAPE combined with EP more effectively induced apoptosis. The number of apoptotic cells in the samples treated with EP and CAPE was significantly higher. The percentage of apoptotic cells also increased with increasing CAPE dose (Figure 6).

## 4. Conclusions

A combination of electroporation and CAPE application efficiently reduces the time required to induce a cytotoxic effect in Me45 and MeWo melanoma cells. This high cytotoxicity correlates with the induction of high oxidative stress and disruption of the metabolism of melanoma cells by inducing apoptotic cell death.

Electroporation of melanoma cells may be an efficient CAPE delivery system, facilitating the application of this compound with a reduction in its dose and exposure time.

## Figures and Tables

**Figure 1 pharmaceutics-12-00478-f001:**
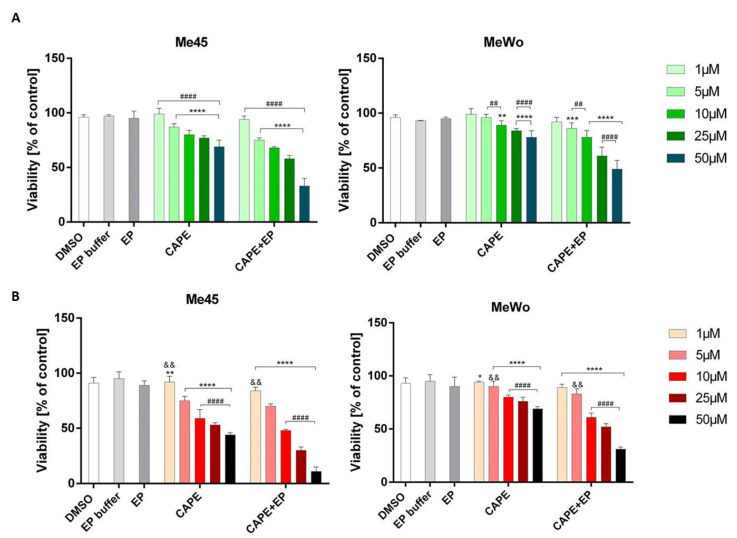
The viability of Me45 and MeWo cell lines after 24 h (**A**) and 72 h (**B**) of incubation following increasing concentrations of caffeic acid phenethyl ester (CAPE) or CAPE connected with electroporation (EP). Viability is expressed as the percentage of control cells (no treated cells). Error bars shown are means ± SD for *n* = 6. For Dunnett’s multiple comparisons tests: * statistically significant for *p* = 0.013, ** statistically significant for *p =* 0.0022, *** statistically significant for *p =* 0.0002, **** statistically significant for *p <* 0.0001. For Sidak’s multiple comparisons tests: ^&&^ statistically significant for *p =* 0.0032, ^##^ statistically significant for *p = 0.0028*, ^####^ statistically significant for *p <* 0.0001.

**Figure 2 pharmaceutics-12-00478-f002:**
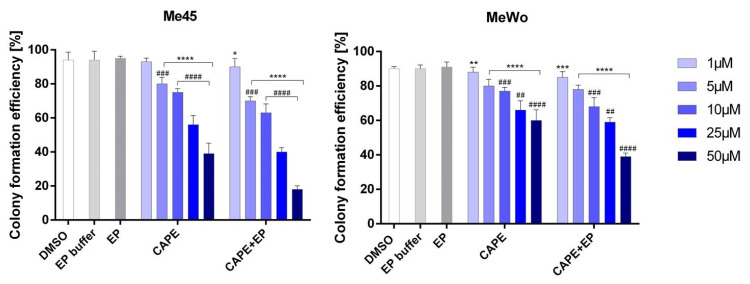
The viability of Me45 and MeWo cell line determined by cloning efficacy test after 11 days. Error bars shown are means ± SD for *n* = 6. For Dunnett’s multiple comparisons tests: * statistically significant for *p* = 0.0178, ** statistically significant for *p* = 0.0031, *** statistically significant for *p* = 0.0003, **** statistically significant for *p <* 0.0001. For Sidak’s multiple comparisons tests: ^##^ statistically significant for *p* = 0.005, ^###^ statistically significant for *p* = 0.0002, ^####^ statistically significant for *p* < 0.0001.

**Figure 3 pharmaceutics-12-00478-f003:**
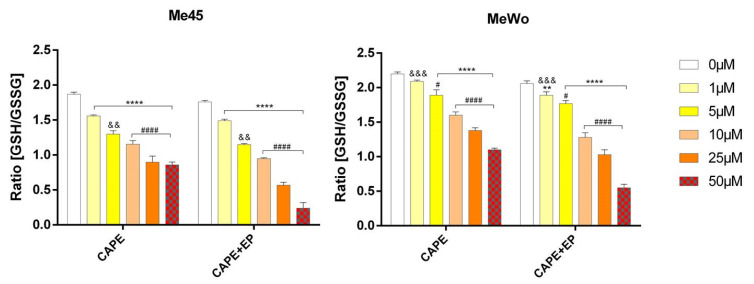
The level of the ratio of reduced to oxidized glutathione (GSH/GSSG) in Me45 and MeWo cell lines after 24-h incubation following increasing concentrations of CAPE or CAPE connected with EP. Error bars shown are means ± SD for *n* = 6. For Dunnett’s multiple comparisons tests: ** statistically significant for *p* = 0.0016, **** statistically significant for *p* < 0.0001. For Sidak’s multiple comparisons tests: ^&&^ statistically significant for *p* = 0.0032, ^&&&^ statistically significant for *p* = 0.0003, ^#^ statistically significant for *p* = 0.043, ^####^ statistically significant for *p* < 0.0001.

**Figure 4 pharmaceutics-12-00478-f004:**
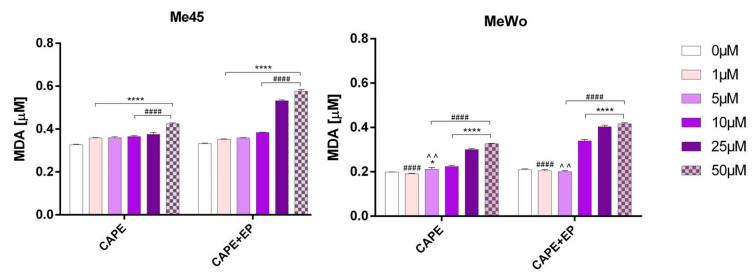
The level of the malondialdehyde (MDA) in Me45 and MeWo cell lines after 24-h incubation following increasing concentrations of CAPE or CAPE connected with EP. Error bars shown are means ± SD for *n* = 6. For Dunnett’s multiple comparisons tests: * statistically significant for *p* = 0.0142, **** statistically significant for *p* < 0.0001. For Sidak’s multiple comparisons tests: ^^^^ statistically significant for *p* = 0.0023, ^####^ statistically significant for *p* < 0.0001.

**Figure 5 pharmaceutics-12-00478-f005:**
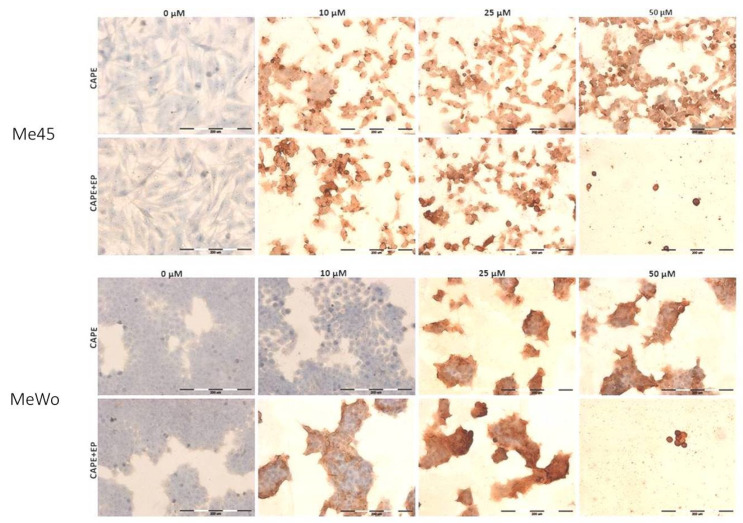
The immunocytochemistry analysis of cleaved PARP-1 in Me45 and MeWo cell lines after 6-h exposure to CAPE or CAPE connected with EP. Results are presented in the Table 1 as the percentage of positively stained cells. The evaluation of stained reaction: (−) negative, no reaction; (+) weak, (++) moderate, and (+++) strong. Results are presented as the mean number of cells counted from three fields (×400).

**Figure 6 pharmaceutics-12-00478-f006:**
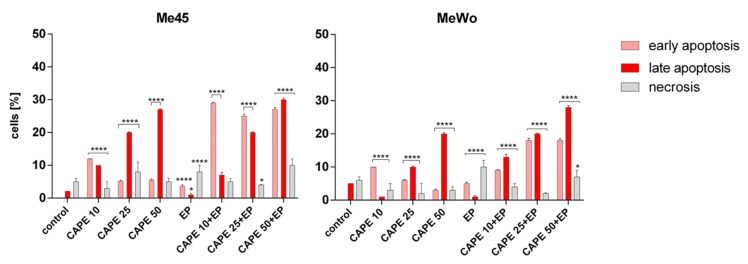
The evaluation of the type of induced cell death by flow cytometry in Me45 and MeWo cells after 6-h incubation with increasing concentrations of CAPE and CAPE combined with EP. Control cells were incubated in cell cultured medium. Error bars shown are means ± SD for *n* = 3; * statistically significant for *p =* 0.0263, **** statistically significant for *p* < 0.0001.

**Table 1 pharmaceutics-12-00478-t001:** The effect of CAPE and CAPE combined to EP on the cleaved-PARP-1 expression in Me45 and MeWo cells.

		The Intensity of Staining		Positively Stained Cells [%]
**Me45**	**CAPE**	CONTROL	-	0
		10 μM	++/+++	100
		25 μM	++/+++	100
		50 μM	+++	100
	**CAPE+EP**	CONTROL	-	0
		10 μM	+++	100
		25 μM	+++	100
		50 μM	+++	100
**MeWo**	**CAPE**	CONTROL	-	0
		10 μM	-	0
		25 μM	+++	100
		50 μM	+++	100
	**CAPE+EP**	CONTROL	-	0
		10 μM	++/+++	100
		25 μM	+++	100
		50 μM	+++	100

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
