# Peer review of "Caffeic Acid Phenethyl Ester Assisted by Reversible Electroporation—In Vitro Study on Human Melanoma Cells"

_pharmaceutics, 2020, doi:10.3390/pharmaceutics12050478_

Round 1

Reviewer 1 Report

The aim of the study entitled “Caffeic acid phenethyl ester assisted by reversible electroporation - in vitro study on human melanoma cells” was to investigate and compare the effectiveness of CAPE and CAPE connected with electroporation against melanoma cells (Me45 and MeWo cell lines).

Melanoma is one of the most serious skin cancers, and it is far more dangerous than others because of its ability to spread to other organs more rapidly if it not treated at an early stage, mainly by surgical intervention.

That is why there is observed a continues research on more effective ways in skin melanoma treatment.

Nowadays, there are natural agents acting as promising elements in chemotherapy and chemoprevention.

One of confirmed but still explored agent is caffeic acid phenethyl ester (CAPE), natural polyphenol extracted from bee propolis. There are research, which study its acting mechanisms in vitro and in vivo, nevertheless all the investigations increasing its activity is justified. It is confirmed, that the cellular absorption of a transported drug can be enhanced by local application of electric pulses to the cancer tissue.

Generally, it is good reading, the construction of the experiments are right; however I would have few concerns mainly related to way of results presenting and interpretation.

References:

In line 38 it is worth to add some more references (not limited to those):

https://doi.org/10.3390/nu9101144

https://doi.org/10.1177/1534735417753545

https://doi.org/10.1080/10520295.2019.1589574

https://doi.org/10.3390/molecules22071124

In Materials and Methods:

  • please indicate, how many experiments (repeats or multiplications) were conducted for every method;
  • update the statistical methods (reasons will be stated below).

Results:

Figure 1:

  1. Which statistic test shows that CAPE+EP works better than CAPE alone for respective doses? What is the p value for that? Dunnet’s test compares the pairs to control only.

Please add one more test comparing all respective doses with CAPE only with CAPE+EP.

Authors may wish the answer if the EP is valuable addition to treat the melanoma cells with CAPE, I assume that is the aim of the study.

Line 102-104 – this conclusion is strictly related to above.

Figure 2, 3, 4. The same problem.

Author Response

Dear Reviewer,

We sincerely appreciate your substantial impact on our contribution and giving us an opportunity to improve our manuscript in the best possible way. According to your suggestions, we have improved the manuscript.

  1. In line 38 it is worth to add some more references (not limited to those): https://doi.org/10.3390/nu9101144;

https://doi.org/10.1177/1534735417753545; https://doi.org/10.1080/10520295.2019.1589574; https://doi.org/10.3390/molecules22071124

Thank you for the good suggestion. We have completed the manuscript with that valuable literature references.

  1. In Materials and Methods: please indicate, how many experiments (repeats or multiplications) were conducted for every method;

Thank you for the valuable comment. We have included in the Materials and Methods section this information.

  1. Update the statistical methods. Please add one more test comparing all respective doses with CAPE only with CAPE+EP.

Thank you for the significant remark. We performed the second statistical test - Sidak's multiple comparisons tests to compare the effect of different concentrations of CAPE to all respective doses of CAPE connected to EP.

We are very grateful for appreciated review process and constructive suggestions. During the review process our manuscript significantly improved and increased more scientific value.

Reviewer 2 Report

Having read the manuscript "Caffeic acid phenethyl ester assisted by reversible electroporation - in vitro study on human melanoma cells" I have the following comments:

  1.  The melanoma cell lines chosen are not matched, MeWo is a BRAFWT cell from a primary melanoma, while Me45 is a BRAFV600E cell from a secondary melanoma.  In light of which any comparison between the two cells is of concern due to their different BRAF status.  Can the authors comment on this, and if possible repeat experiments using a BRAFWT cell from a secondary melanoma or a BRAFV600E cell taken from a primary melanoma.
  2. Can electroporation be used in vivo and only applied to a group of cells?  What evidence is there to show that this is possible?
  3. Why did the authors not probe for PARP cleavage using relevant antibodies on Western blots?
  4. The materials and methods section is very brief and needs to be rewritten to explain in detail what the authors did.
  5. How many events were measured in the flow cytometry experiments?
  6. In the viability studies what was the final concentration of DMSO in the solutions?
  7. Line 256 what is the ABC method, please explain.
  8. Explain what is the composition of the EP buffer?
  9. What is the relationship, if any, is there between GSH/GSSG vs MDA in these cells?
  10. The authors need to rewrite sections of the manuscript as it is unclear as to what they mean.  Having an native English speaker assist them would greatly assist in this revision

Author Response

Dear Reviewer,

We sincerely appreciate your efforts that you put in  our contribution and giving us another opportunity to improve our manuscript in the best possible way. According to your suggestion, we have improved the manuscript.

  1. The melanoma cell lines chosen are not matched, MeWo is a BRAFWTcell from a primary melanoma, while Me45 is a BRAFV600Ecell from a secondary melanoma.  In light of which any comparison between the two cells is of concern due to their different BRAF status.  Can the authors comment on this, and if possible repeat experiments using a BRAFWT cell from a secondary melanoma or a BRAFV600E cell taken from a primary melanoma.

Thank you very much for this interesting suggestion. Our study intended to investigate the effect of CAPE in combination with EP on melanoma cells. By using a physical factor through which we induced the formation of transient pores in the cell membrane, we focused on assessing the mechanisms induced in the cell without the participation of signaling pathways determined by the activation of receptors located on the surface of the cell membrane. For this reason, we did not analyze the melanoma cell lines selected for experiment for BRAF status. It is also worth to remember that activation of the mitogen-activated protein kinase pathway in malignant melanoma can occur independently of the BRAF mutation [Yazdi AS et al. 2010], and the mitogen-activated protein kinase activity is a subject of regulation even in BRAF/NRAS mutant melanoma cells and the high MAPK pathway signaling may be important only in distinct subsets of melanoma cells [Houben R et al. 2008]. Nevertheless, the suggested suggestion is interesting and we will gladly deal with this problem in subsequent studies. The above manuscript is reported as Communication, and therefore has volume restrictions.

Yazdi AS, et al. Activation of the mitogen-activated protein kinase pathway in malignant melanoma can occur independently of the BRAF T1799A mutation. Eur J Dermatol. 2010, 20(5):575-9.

Houben R, et al. Phospho-ERK staining is a poor indicator of the mutational status of BRAF and NRAS in human melanoma. J Invest Dermatol. 2008, 128(8):2003-12.

  1. Can electroporation be used in vivoand only applied to a group of cells?  What evidence is there to show that this is possible?

Thank you for that question. The electroporation phenomenon became a more popular technique for loading cells with molecules that are either not possible or difficult to pass through the cells. This directed to the growth of EP-based technology for biomedical applications and researches in the field of drug delivery and gene therapy, like gene electrotransfection, nonthermal irreversible electroporation, and electrochemotherapy (ECT) [MiklavÄŤiÄŤ D, et al. 2012, Groselj A, et al. 2015]. ECT is a local and nonthermal tumor ablation modality, which combines the administration of a poorly permeant cytotoxic agent with the local application of electric pulses that induce reversible EP, thus improving drug diffusion into the cells. Through this method, the efficacy of chemotherapeutic medications increases by the use of electrical pulses which provides good local tumor control [Cabula C, et al, 2015, Sersa G, et al., 2012].

ECT effectiveness has been approved in different types of tumors. The first clinical study was published in 1991 on head and neck tumor nodules. ECT has been used in the treatment of subcutaneous and cutaneous lesions and metastases from tumors, with objective response ranging from 75% to 99%. It is applied for treating melanomas, sarcomas, and other types of skin cancer, cervix leiomyosarcoma, and breast cancer [Kunte C, et al. 2017, Mir LM, et al. 1991]. ECT has been known by some of the national health services for the treatment of patients with cutaneous metastases from various tumor histotypes. It has been adapted over 150 cancer centers throughout Europe and has contributed to its diffusion in clinical practice [Esmaeili N, et al. 2019].

Falk H. et al. reported a spectacular description of a clinical case of melanoma ECT [Falk H, et al. 2017]. That article demonstrates that electrochemotherapy is a suitable treatment for fragile, elderly patients. In this case, the 83 years old patient that was diagnosed and had developed comorbidities disease. Due to comorbidity, the patient did not tolerate chemotherapy very well. When referred for electrochemotherapy, her general condition was poor, but her toleration of the treatment was acceptable, with minimal hospitalization (4 days). Below, there is the Figure from that article, showing the patient's response to treatment in time.

Falk et al. 2017 (Figure attached in PDF version of the response to the review).

MiklavÄŤiÄŤ D, at el. Electrochemotherapy: technological advancements for efficient electroporation-based treatment of internal tumors. Medical and Biological Engineering and Computing. 2012, 50, 1213-1225. 2012.

Groselj A, et al. Coupling treatment planning with navigation system: a new technological approach in treatment of head and neck tumors by electrochemotherapy. BioMedical Engineering OnLine. 2015, 14, 2.

Cabula C, et al. Electrochemotherapy in the treatment of cutaneous metastases from breast cancer: a multicenter cohort analysis.  Annals of Surgical Oncology. 2015, 22, 442-450.

Sersa G, et al. Electrochemotherapy of chest wall breast cancer recurrence. Cancer Treatment Reviews. 2012, 38, 379-386.

Kunte C, et al. Electrochemotherapy in the treatment of metastatic malignant melanoma: a prospective cohort study by InspECT. British Journal of Dermatology. 2017, 176, 1475-1485.

Mir LM, et al. Electrochemotherapy, a new antitumor treatment: first clinical trial. Comptes Rendus de l’Academie des Sciences. Serie III, Sciences de la vie. 1991, 313, 613-618.

Esmaeili N, et al. Electrochemotherapy: A Review of Current Status, Alternative IGP Approaches, and Future Perspectives. J Healthc Eng. 2019, 3, 2784516.

Falk H, et al. Electrochemotherapy and calcium electroporation inducing a systemic immune response with local and distant remission of tumors in a patient with malignant melanoma - a case report. Acta Oncol. 2017, 56, 1126-1131.

  1. Why did the authors not probe for PARP cleavage using relevant antibodies on Western blots?

Thank you for the question. We decided to evaluate the cleaved-PARP protein by the ICC method because we also wanted to observe cell morphology after the therapeutic method used. We will examine it by western blot in our further investigations. Here we have a preliminary study that are a basis for the extended research. The above manuscript is submitted as  ”Communication", further research will be continued.

  1. The materials and methods section is very brief and needs to be rewritten to explain in detail what the authors did.

       Thank You for that suggestion. The section Materials and Methods was developed in the revised manuscript.

  1. How many events were measured in the flow cytometry experiments?

       All samples in the flow cytometry experiment were analyzed in triplicate, in case of each sample minimum 10,000 events were analyzed. This information has been included in the revised manuscript.

  1. In the viability studies what was the final concentration of DMSO in the solutions?

DMSO concentration in the samples with the highest tested CAPE solution was 0.75 mM. At the lower doses of CAPE, DMSO concentrations were correspondingly lower. As was shown in publications on the in vitro study of DMSO cytotoxicity, doses in the range of 0.05-1 mM do not cause a cytotoxic effect, on the contrary, they slightly stimulate cell proliferation [Hebling, et al. 2015].

Hebling, J., et al. Cytotoxicity of dimethyl sulfoxide (DMSO) in direct contact with odontoblast-like cells. Dental Materials, 2015, 31, 399-405.

  1. Line 256 what is the ABC method, please explain.

A detailed description of the method has been provided in the revised manuscript:

“Immunocytochemistry was performed after 6 hours of incubation using the Avidin-Biotin Complex (ABC) method. It is a standard and one of the extensively used technique for immunocytochemistry staining. Avidin, which is a large glycoprotein, can be labeled with peroxidase and has a very high affinity for biotin. That low molecular weight vitamin is used in that method to form conjugation with antibodies. Immunohistochemical staining intensity is a function of the enzyme activity, and enhanced sensitivity can be obtained by expanding the number of enzyme molecules bound to the antigen. The multiplication binding opportunities between the tetravalent avidin and biotinylated antibodies (connected to the antigen) are the idea for obtaining clear marked binding site antigen-antibody. The procedure involved the following stages, based on applying three layers: the first layer was an unlabeled primary antibody. The second layer was the biotinylated secondary antibody and the third one was a complex of avidin-biotin-peroxidase. The peroxidase then interacted with diaminobenzidine (DAB) and it resulted in creating colorimetric end products. Fixed melanoma cells after CAPE or CAPE + EP treatment were stained. The procedure was performed on basic microscope slides. Anti-Cleaved PARP1 antibody (Abcam, ab32561) was used as the primary antibody using a 1:100 dilution. Other reagents were from the staining kit (Dako REAL EnVision Detection System, Peroxidase / DAB +, Rabbit / Mouse, K5007, Agilent, USA). Cell nuclei were stained with hematoxylin (Roth, Germany). The microscope slides were examined with the upright microscope (Olympus BX51, Japan). Staining assessments were determined by counting 100 cells in randomly chosen fields. The staining was estimated positive if it was discerned in more than 5% of cells. The intensity of staining was assessed as (-) negative, (+) weak, (++) moderate, and (+++) strong”.

  1. Explain what is the composition of the EP buffer?

During electroporation, procedure cells were suspended in buffer with low electrical conductivity (10 mM KH2PO4/K2HPO4, 1 mM MgCl2, 250 mM sucrose, pH 7.4). This information has been included in the manuscript.

  1. What is the relationship, if any, is there between GSH/GSSG vs MDA in these cells?

Oxidative stress is caused due to an imbalance between production of reactive oxygen species (free radicals) and effectiveness of antioxidant defense. Reactive oxygen species (ROS) play a crucial role in cell signaling, however when the balance between ROS production and consumption is disrupted, it can lead to cell damage. Oxidative stress can cause damage to DNA, proteins and lipids. Glutathione exist in reduced (GSH) and oxidized (GSSG) forms in cells and tissues. provides reducing equivalents to antioxidant enzymes, hydroxyl radicals, ROS and is itself oxidized to GSSG; therefore GSH/GSSG ratio is critical indicator of the health of cell. During oxidative stress there is observed a decrease in levels of GSH and an increase in levels of GSSG and thus GSH/GSSG ratio decreases.

MDA (malondialdehyde) is the most commonly used biomarker of oxidative stress in the cell, resulting from damage to membrane lipids. MDA is a secondary product generated during the reaction from the decay of products of lipid peroxidation.

There is no direct relationship between GSH/GSSG and MDA. These are two different markers of oxidative stress in the cell. Assessing them, we wanted to show that oxidative stress-induced in cells causes two types of pathological changes. These are modifications in the membrane lipids and modifications of proteins (thiol groups damage) involved in the neutralization of oxygen free radicals.

  1. The authors need to rewrite sections of the manuscript as it is unclear as to what they mean. Having an native English speaker assist them would greatly assist in this revision.

       The manuscript has been subjected to language correction by native speaker.

We are very grateful for appreciated review process and constructive suggestions. During the review process our manuscript significantly improved and increased more scientific value.                          

Reviewer 3 Report

In this study, the effects of caffeic acid phenylethyl ester (CAPE), a natural phenolic compound known for its antioxidative, immunomodulatory and (dose-dependent) cytotoxic properties, on two melanoma cell lines in conjunction with electroporation (EP) were analyzed in vitro.

Following up on previous investigations showing (selective) negative/cytotoxic effects of CAPE on melanoma and tumor cell viability in general, the authors in their present manuscript specifically tested the combined effect of CAPE exposure in combination with EP vs. DMSO on primary (MeWo) vs. metastatic (Me45) melanoma cell line – generating new results with regard to the efficacy of electroporation as “CAPE-delivery method”. As read-outs for the effect of CAPE+/-EP on melanoma cells, MTT assay, cloning assays, oxidative stress/lipid peroxidation, IHC for cleaved PARP and Annexin V staining by FACS were used. Overall the study presents valid results and addresses the interesting question if and how electroporation may enhance the toxic and pro-apoptotic effects of phenolic compounds on melanoma cell lines. However, due to the already relatively well know effects of CAPE on tumors, a few additional mechanistic insights may improve the study.

General points:

1. It would be helpful if the authors provided the BRAF-status of the cell lines used at some point (maybe it is mentioned, but I had to look it up online as it was not explicitly mentioned at the beginning of the results or in the Materials&Methods). Could the authors maybe even elaborate and discuss more on how the kinase status of the melanoma cells may be important with regard to the action of CAPE, GST/redox and apoptosis?

2. Could the authors explain comprehensively why normally, CAPE is known to be antioxidative and cell protective, but in melanoma or tumor cells, rather has pro-oxidative and pro-apoptotic effects – e.g. what are the doses and to what extent does the conversion in toxic compounds take place intracellularly/in the human body? Would they expect adverse effects or complementary effects – e.g. on the niche or on immune cells? How would they apply EP in patients? – e.g. skin metastases? Could they discuss the effect of CAPE+/- EP in context with current targeted therapy approaches and other delivery methods (e.g. Chinembiri et al, Molecules 2014; Gatzka et al., IJMS 2018; Clemente et al., IJMS 2018)

3. As the authors refer to the necessity of tyrosinase in melanoma cells to generate the pro-oxidative quinone/GST depletion, - is quinone only produced in melanoma cells or also in other tumor cells? Have the authors measured tyrosinase activity (or levels) or tried to knock-down tyrosinase or to use amelanotic melanoma cells? Have the authors tried non-melanoma cells and non-tumor cells - and may EP improve CAPE cytotoxicity as well? (tyrosinase – GST/Quinone derivate pro-oxidative?)

4. Specific questions
Figure 1+2: Why are metastatic cells more sensitive? If it was due to different tyrosinase levels –have the authors had a chance to measure tyrosinase levels or activity or to knock down tyrosinase?
- Was DMSO used as dissolvent and at what final concentration? Usually, also DMSO + EP may have toxic effects as well, but it is not shown.
- Was the intracellular concentration of CAPE, CAPE-SG or CAPE-quinone increased after CAPE + EP compared with CAPE without EP - or could there be any potentially synergistics effect due to minor membrane damage by EP ? Could the authors measure the intracellular concentration of CAPE or the conjugates after EP or without EP?

Figures 3-6: Mechanistically, increased Annexin V by FACS and cleaved PARP by IHC were measured indicating increased apoptosis in conjunction with increased ROS – could the authors present or follow up with an analysis of pro-apoptotic proteins such as Bim/Bad/Puma or the NF-kB pathway analysis as there is crosstalk between oxidative stress/GST and NF-kB signalling? How is the effect on redox balance modulated by EP?

A graphical illustration/summary of effects CAPE and CAPE+EP in melanoma cells would be helpful.

Multiple typos and syntax errors – authors should have their manuscript checked by native English speaker.

Author Response

Dear Reviewer,

We sincerely appreciate your significant contribution in the revision of our manuscript and giving us another opportunity to improve our manuscript in the best possible way. According to your suggestion, we have improved the manuscript.

  1. It would be helpful if the authors provided the BRAF-status of the cell lines used at some point (maybe it is mentioned, but I had to look it up online as it was not explicitly mentioned at the beginning of the results or in the Materials&Methods). Could the authors maybe even elaborate and discuss more on how the kinase status of the melanoma cells may be important with regard to the action of CAPE, GST/redox and apoptosis?

Thank you very much for this suggestion. MeWo is a BRAFWT melanoma cell line, while Me45 is a BRAFV600E melanoma cell line. This information has been included in the revised manuscript. Our study intended to investigate the effect of CAPE in combination with EP on melanoma cells. By using a physical factor through which we induced the formation of transient pores in the cell membrane, we focused on assessing the mechanisms induced in the cell without the participation of signaling pathways determined by the activation of receptors located on the surface of the cell membrane. For this reason, we did not analyze the melanoma cell lines selected for experiment for BRAF status. It is also worth remembering that activation of the mitogen-activated protein kinase pathway in malignant melanoma can occur independently of the BRAF mutation [Yazdi AS et al. 2010], and the mitogen-activated protein kinase activity is subject to regulation even in BRAF/NRAS mutant melanoma cells and the high MAPK pathway signaling may be important only in distinct subsets of melanoma cells [Houben R et al. 2008]. Nevertheless, the Reviewer’s suggestion is interesting and we will gladly deal with this problem in subsequent studies. The above manuscript is reported as Communication, and therefore has volume restrictions.

Yazdi AS, et al. Activation of the mitogen-activated protein kinase pathway in malignant melanoma can occur independently of the BRAF T1799A mutation. Eur J Dermatol. 2010, 20(5):575-9.

Houben R, et al. Phospho-ERK staining is a poor indicator of the mutational status of BRAF and NRAS in human melanoma. J Invest Dermatol. 2008, 128(8):2003-12.

  1. Could the authors explain comprehensively why normally, CAPE is known to be antioxidative and cell protective, but in melanoma or tumor cells, rather has pro-oxidative and pro-apoptotic effects – e.g. what are the doses and to what extent does the conversion in toxic compounds take place intracellularly/in the human body? Would they expect adverse effects or complementary effects – e.g. on the niche or on immune cells? How would they apply EP in patients? – e.g. skin metastases? Could they discuss the effect of CAPE+/- EP in context with current targeted therapy approaches and other delivery methods (e.g. Chinembiri et al, Molecules 2014; Gatzka et al., IJMS 2018; Clemente et al., IJMS 2018).

Thank you for those questions. CAPE has strong antioxidant properties, which is due to its chemical structure. Its different activity concerning melanoma cells results from the specific metabolic conditions of these cells. Tyrosinase overexpression in melanoma cells is crucial in this aspect. The bioactivation of CAPE by tyrosinase to its corresponding o-quinone metabolite would lead to selective melanoma cytotoxicity. The toxic metabolite o-quinone is capable of reactive oxygen species formation and intracellular GSH depletion in melanoma cells [Kudugunti et al. 2011] (Scheme 1). 

Scheme 1. Kudugunti et al. 2011 (Scheme attached in PDF version of the response to the review).

Among all resistance mechanisms involved in melanoma therapy, the overexpression of GST and MRP may play critical roles [Kudugunti et al. 2010]. Because tyrosinase is overexpressed in melanoma, this may also enable selective inhibition of GST as a secondary target in melanoma cells compared to non-melanoma cells that do not express tyrosinase. Bioactivated CAPE and its GSH conjugates selectively inhibit GST in the presence of tyrosinase [Kudugunti et al. 2011].

As a phytotherapeutic , CAPE has been shown to induce three types of antitumor activity in cancer cells – anti-proliferative, anti-metastatic, and apoptosis induction [Premratanachai et al. 2014; Wadhwa et al. 2016] and although it has no cytotoxic activity against normal cells. There are some possibilities that as a herbal medicinal product it can be toxic if  accumulatds beyond the acceptable level in the human body. However, within this concept we have presented, CAPE is selectively delivered to cells through transitional pores, formed after electroporation, and we minimize this problem. The electroporation phenomenon became a more popular technique for loading cells with active molecules that are either not possible or difficult to pass through the cell membranes. This directed to spreading  EP-based technologies for biomedical applications, drug delivery and gene therapy, like gene electrotransfection, nonthermal irreversible electroporation, and electrochemotherapy (ECT) [MiklavÄŤiÄŤ D, et al. 2012, Groselj A, et al. 2015]. ECT is a local and nonthermal tumor ablation modality, which combines the administration of a poorly permeant cytotoxic agent with the local application of electric pulses that induce reversible EP, thus improving drug diffusion into the cells. Through this method, the efficacy of chemotherapeutic medications increases by the use of electrical pulses which provides good local tumor control [Cabula C, et al, 2015, Sersa G, et al., 2012].

ECT effectiveness has been approved in different types of tumors. The first clinical study was published in 1991 on head and neck tumor nodules. ECT has been used in the treatment of subcutaneous and cutaneous lesions and metastases from tumors, with objective response ranging from 75% to 99%. It is applied for treating melanomas, sarcomas, and other types of skin cancer, cervix leiomyosarcoma, and breast cancer [Kunte C, et al. 2017, Mir LM, et al. 1991]. ECT has been known by some of the national health services for the treatment of patients with cutaneous metastases from various tumor histotypes. It has been adapted over 150 cancer centers throughout Europe and has contributed to its diffusion in clinical practice [Esmaeili N, et al. 2019].

Falk H. et al. reported a spectacular description of a clinical case of melanoma ECT [Falk H, et al. 2017]. That article demonstrates how electrochemotherapy is a suitable treatment for fragile, elderly patients. In this case, the patient was 83 years old at diagnosis and had developed comorbidities disease. Due to comorbidity, she did not tolerate chemotherapy very well. When referred for electrochemotherapy, her general condition was poor, but her toleration of the treatment was acceptable, with minimal hospitalization (4 days). Below, there is the Figure from that article, showing the patient's response to treatment in time.

Falk et al. 2017 (Figure attached in PDF version of the response to the review).

Kudugunti, S.K.; Vad, N.M.; Ekogbo, E.; Moridani, M.Y. Efficacy of caffeic acid phenethyl ester (CAPE) in skin B16-F0 melanoma tumor bearing C57BL/6 mice. Invest. New Drugs 2011, 29, 52-62.

Kudugunti, S.K.; Vad, N.M.; Whiteside, A.J.; Naik, B.U.; Yusuf, M.A.; Sirvenugpol, K.S.; Moridani, M.Y. Biochemical mechanism of caffeic acid phenylethyl ester (CAPE) selective toxicity towards melanoma cell lines. Chem. Biol. Interact. 2010, 188, 1-14.

Premratanachai, P. and Chanchao C. Review of the anticancer activities of bee products. Asian Pac J Trop Biomed. 2014 May; 4(5): 337–344.

Wadhwa, R., Nigam, N., Bhargava, P., et al. Molecular Characterization and Enhancement of Anticancer Activity of Caffeic Acid Phenethyl Ester by γ Cyclodextrin. J Cancer. 2016; 7(13): 1755–1771.

MiklavÄŤiÄŤ D, at el. Electrochemotherapy: technological advancements for efficient electroporation-based treatment of internal tumors. Medical and Biological Engineering and Computing. 2012, 50, 1213-1225. 2012.

Groselj A, et al. Coupling treatment planning with navigation system: a new technological approach in treatment of head and neck tumors by electrochemotherapy. BioMedical Engineering OnLine. 2015, 14, 2.

Cabula C, et al. Electrochemotherapy in the treatment of cutaneous metastases from breast cancer: a multicenter cohort analysis.  Annals of Surgical Oncology. 2015, 22, 442-450.

Sersa G, et al. Electrochemotherapy of chest wall breast cancer recurrence. Cancer Treatment Reviews. 2012, 38, 379-386.

Kunte C, et al. Electrochemotherapy in the treatment of metastatic malignant melanoma: a prospective cohort study by InspECT. British Journal of Dermatology. 2017, 176, 1475-1485.

Mir LM, et al. Electrochemotherapy, a new antitumor treatment: first clinical trial. Comptes Rendus de l’Academie des Sciences. Serie III, Sciences de la vie. 1991, 313, 613-618.

Esmaeili N, et al. Electrochemotherapy: A Review of Current Status, Alternative IGP Approaches, and Future Perspectives. J Healthc Eng. 2019, 3, 2784516.

Falk H, et al. Electrochemotherapy and calcium electroporation inducing a systemic immune response with local and distant remission of tumors in a patient with malignant melanoma - a case report. Acta Oncol. 2017, 56, 1126-1131.

  1. As the authors refer to the necessity of tyrosinase in melanoma cells to generate the pro-oxidative quinone/GST depletion, - is quinone only produced in melanoma cells or also in other tumor cells? Have the authors measured tyrosinase activity (or levels) or tried to knock-down tyrosinase or to use amelanotic melanoma cells? Have the authors tried non-melanoma cells and non-tumor cells - and may EP improve CAPE cytotoxicity as well? (tyrosinase – GST/Quinone derivate pro-oxidative?)

Thank you for the suggestion. Melanomas tyrosinase, the main enzymes of melanogenesis is overexpressed in melanotic melanoma cells. Loss of pigmentation in amelanotic melanoma is common in advanced lesions because of dysfunction in melanogenesis proteins: tyrosinase and tyrosinase-related proteins 1 and 2. In our preliminary studies we focused mainly on the CAPE action on the tyrosinase in melanogenesis pathway and inhibition of glutathione transferase [Kudugunti SK et al.].  We didn’t check the expression of tyrosinase in amelanotic cancer cells, but, it is a good idea to control it in our further investigations. The revised manuscript is submitted as a ”Communication", further research will be continued.

We didn’t focus on normal and other cancer cells because of many investigations of antiproliferative examinations of CAPE. Additionally, there are many evidences, which elaborate the antiproliferation activity of CAPE in various type of cancer in different kind of pathways [Firat et al. 2019].

Fırat, F., Özgül, M., Türköz Uluer, E. et al. Effects of caffeic acid phenethyl ester (CAPE) on angiogenesis, apoptosis and oxidatıve stress ın various cancer cell lines. Biotech Histochem. 2019, 94(7):491-497.

Kudugunti, S.K.; Vad, N.M.; Ekogbo, E.; Moridani, M.Y. Efficacy of caffeic acid phenethyl ester (CAPE) in skin B16-F0 melanoma tumor bearing C57BL/6 mice. Invest. New Drugs 2011, 29, 52-62.

  1. Figure 1+2: Why are metastatic cells more sensitive? If it was due to different tyrosinase levels –have the authors had a chance to measure tyrosinase levels or activity or to knock down tyrosinase?

       Further experience is needed to answer this question. The assessment of the level of active tyrosinase in the melanoma cells is a very good suggestion and it will be checked in the next stages of the study.

  1. Was DMSO used as dissolvent and at what final concentration? Usually, also DMSO + EP may have toxic effects as well, but it is not shown.

DMSO concentration in the samples with the highest tested CAPE solution was 0.75 mM. At the lower doses of CAPE, DMSO concentrations were correspondingly lower. As shown in publications on the in vitro study of DMSO cytotoxicity, doses in the range of 0.05-1 mM do not cause a cytotoxic effect, on the contrary, they slightly stimulate cell proliferation [Hebling, et al. 2015]. The addition of DMSO to this electroporation buffer improves the efficiency of the electroporation process and also has a slight effect on increasing cell proliferation [Melkonyan, et al. 1996].

Melkonyan, H. et al. Electroporation Efficiency in Mammalian Cells is increased by Dimethyl Sulfoxide (DMSO). Nucleic Acids Research, 1996, 24, 4356-4357.

Hebling, J., et al. Cytotoxicity of dimethyl sulfoxide (DMSO) in direct contact with odontoblast-like cells. Dental Materials, 2015, 31, 399-405.

  1. Was the intracellular concentration of CAPE, CAPE-SG or CAPE-quinone increased after CAPE + EP compared with CAPE without EP - or could there be any potentially synergistics effect due to minor membrane damage by EP ? Could the authors measure the intracellular concentration of CAPE or the conjugates after EP or without EP?

The electroporation parameters used in our study  did not cause cytotoxic effect and oxidative stress in the cells. The use of electroporation to deliver CAPE to cells is intended to increase the efficiency of CAPE delivery into the cells, and thus if we deliver more CAPE, then a larger amount of CAPE-SG or CAPE-quinone will be created in the cell. This assumption is consistent with our results. The evaluation of the CAPE and its conjugates after EP and without EP is an interesting suggestion. We will gladly undertake that point in our future research.

  1. Figures 3-6: Mechanistically, increased Annexin V by FACS and cleaved PARP by IHC were measured indicating increased apoptosis in conjunction with increased ROS – could the authors present or follow up with an analysis of pro-apoptotic proteins such as Bim/Bad/Puma or the NF-kB pathway analysis as there is crosstalk between oxidative stress/GST and NF-kB signalling? How is the effect on redox balance modulated by EP?

I completely agree with that suggestion. In order to develop a new therapeutic method, it is necessary to thoroughly understand the pathways induced by it in cells. This is currently the subject of our research and will be the subject of another extended manuscript. This article is limited by volume due to the Communication formula.

  1. A graphical illustration/summary of effects CAPE and CAPE+EP in melanoma cells would be helpful.

The graphic abstract has been attached to the revised manuscript.

  1. Multiple typos and syntax errors – authors should have their manuscript checked by native English speaker.

       The manuscript has been subjected to language correction.

We are very grateful for appreciated review process and constructive suggestions. During the review process our manuscript significantly improved and increased more scientific value.              

Round 2

Reviewer 2 Report

Having read the revised manuscript as well as the rebuttal letter, I have no further comments on the manuscript.  I look forward to reading about the results from their planned experiments with melanoma cells.

Reviewer 3 Report

The authors have addressed most of my suggestions and questions - and have improved the manuscript accordingly. Some additional references and explanations have been included. Although no additional experiments have been performed, I hope that the authors will get the chance in the nearer future, to further clarify the mechanisms of the combined action of CAPE and EP on different types of melanoma and the tumor environment. The addition of a graphical abstract is appreciated. With the English style corrections, the communication is now a lot better and easier to read. Overall, the manuscript has been improved and I now recommend its acceptance as communication in Pharmaceutics.